# EvoCoT: Overcoming the Exploration Bottleneck in Reinforcement Learning for LLMs

## Abstract

Reinforcement learning with verifiable reward (RLVR) has become a promising paradigm for post-training large language models (LLMs) to improve their reasoning capability. However, when the rollout accuracy is low on hard problems, the reward becomes sparse, limiting learning efficiency and causing exploration bottlenecks. Existing approaches either rely on teacher models for distillation or filter out difficult problems, which limits scalability or restricts reasoning improvement through exploration.

We propose **EvoCoT**, a self-**Evo**lving curriculum learning framework based on two-stage **C**hain-**of**-**T**hought (CoT) reasoning optimization. **EvoCoT** constrains the exploration space by self-generating and verifying CoT trajectories, then gradually shortens CoT steps to expand the space in a controlled way. The framework enables LLMs to stably learn from initially unsolved hard problems under sparse rewards. We apply **EvoCoT** to multiple LLM families, including Qwen, DeepSeek, and Llama. Experiments show that **EvoCoT** enables LLMs to solve previously unsolved problems, improves reasoning capability without external CoT supervision, and is compatible with various RL fine-tuning methods. We release the source code to support future research[1].

## 1 Introduction

Recently, reinforcement learning with verifiable reward (RLVR) has emerged as a promising paradigm for the post-training of large language models (LLMs). LLMs demonstrate remarkable reasoning capability in solving complex tasks, from math problems to code generation. Existing works DeepSeek-AI et al. (2025); Liu et al. (2025b) compute rewards via rule-based verification of predicted final answers, effectively enhancing reasoning capability without relying on annotated reasoning trajectories.

Within RLVR, we expect LLMs to explore correct reasoning trajectories during rollouts to obtain rewards and gradually improve their reasoning capability Yu et al. (2025); Shao et al. (2024). However, when the rollout accuracy is low on some hard problems, the LLM receives sparse rewards, hindering the improvement of reasoning capability. Due to the vast solution space, LLMs often face exploration bottlenecks on such problems.

In experiments, we find that LLMs often fail to sufficiently learn from hard problems, even after RLVR training. For example, when trained on GSM8K Cobbe et al. (2021) and MATH Hendrycks et al. (2021) training sets, Qwen2.5-7B still fails to solve 8.8% and 22.0% of the problems, respectively (see Table 2). These unsolved problems are still valuable for RLVR. If LLMs could exploit such problems more effectively during training, their reasoning capability could be further improved Liu et al. (2025b).

Several recent works attempt to address this question. ❶ One category of methods depends on teacher LLMs to provide hints or reasoning trajectories for **distillation** Nath et al. (2025); Ma et al. (2025); Yan et al. (2025); Wu et al. (2025); Fu et al. (2025). For instance, LUFFY Yan et al. (2025) mixes outputs from teacher LLMs into the GRPO candidate set and applies importance sampling to emphasize low-probability but correct actions. These methods enhance performance but require

---

[1] https://anonymous.4open.science/r/EvoCoT-anonymous-76EB

access to teacher LLMs, which is a strong assumption that imposes high costs and limits scalability, especially when training flagship models without available teacher models. ❷ Another category of methods attempts to control problem difficulty to facilitate curriculum learning for LLMs Chen et al. (2025b); Bae et al. (2025); Shi et al. (2025). RORL Bae et al. (2025) computes the rollout accuracy for each group in a batch and retains only the problems within a predefined accuracy range. While this mitigates reward sparsity, it also **filters out** many hard problems that could serve as valuable training data, restricting the LLM's reasoning improvement through exploration. A detailed comparison is provided in Table 1.

In this paper, we aim to investigate the following question:

> **Key Question**
>
> Can LLMs become self-evolving by overcoming exploration bottlenecks and progressively enhancing reasoning capability, without distillation from teacher models?

Table 1: The comparison between existing reinforcement learnin (RL) methods and **EvoCoT**.

| Methods | ❶ Distillation-Free | ❷ Unfiltered |
|---|:---:|:---:|
| ReLIFT Ma et al. (2025) | ✗ | ✗ |
| AdaRFT Shi et al. (2025) | ✓ | ✗ |
| RORL Bae et al. (2025) | ✓ | ✗ |
| TAPO Wu et al. (2025) | ✗ | ✓ |
| LUFFY Yan et al. (2025) | ✗ | ✓ |
| Guide-GRPO Nath et al. (2025) | ✗ | ✗ |
| SRFT Fu et al. (2025) | ✗ | ✓ |
| **EvoCoT** (Ours) | ✓ | ✓ |

We think that the low rollout accuracy on hard problems is primarily due to the vast solution space being far beyond the LLMs' current reasoning capability, as shown in Figure 1. We propose **Evo-CoT**, a self-**Evo**lving curriculum learning framework based on two-stage **C**hain-**o**f-**T**hought (CoT) reasoning optimization. The core idea of **EvoCoT** is to constrain the size of the exploration space. In Stage 1, the LLM receives problems and final answers, and generates its own CoT trajectories. These CoTs are filtered and verified to construct step-by-step reasoning. In Stage 2, **EvoCoT** performs curriculum learning by progressively removing reasoning steps from each CoT trajectory. This step-wise reduction gradually expands the exploration space in a controlled manner, increasing reasoning difficulty while enabling stable training under sparse rewards. Through self-evolving iterations, the LLM enhances its reasoning capability and generates higher-quality CoTs, progressively solving a portion of initially unsolved hard problems.

We apply **EvoCoT** to LLMs across diverse model families, including Qwen, DeepSeek, Llama, and DeepSeek-R1-Distill-Qwen (referred to as R1-Qwen). Experimental results demonstrate that:

- Compared to GRPO, **EvoCoT** enables LLMs to overcome exploration bottlenecks on previously unsolved training set problems, with average improvements of +4.5 for Qwen2.5-7B and +21.7 for R1-Qwen-1.5B.

- Beyond the training set, **EvoCoT** transfers its learned reasoning to other math benchmarks, outperforming SimpleRL with average improvements of +2.3 on Qwen2.5-7B and +2.1 on R1-Qwen-1.5B.

- Compared to SFT and GRPO, **EvoCoT** supports more effective self-exploration, achieving average improvements of +10.8 and +1.6 across all evaluated LLMs.

## 2 RELATED WORK

### 2.1 REINFORCEMENT LEARNING WITH VERIFIABLE REWARD

RLVR for LLMs has drawn considerable research attention following DeepSeek-R1 DeepSeek-AI et al. (2025) and Kimi-k1.5 Team et al. (2025). However, recent studies Yue et al. (2025); Zhao et al. (2025) suggest that the performance of the RLVR-trained model is fundamentally constrained by the base model's inherent capability, as RLVR only biases the base model's output distribution toward reward-maximizing paths. In RLVR, rewards are sometimes too sparse compared to the large solution space, causing exploration bottlenecks that prevent finding solutions unexplored by the base model. Some works Ma et al. (2025); Chen et al. (2025a); Fu et al. (2025); Liu et al. (2025c); Wu et al. (2025); Yan et al. (2025); Nath et al. (2025); Goldie et al. (2025) attempt to incorporate off-policy data into training. For instance, ReLIFT Ma et al. (2025), SASR Chen et al. (2025a), SRFT Fu et al. (2025) and SuperRL Liu et al. (2025c) integrate RLVR with supervised fine-tuning (SFT). Meanwhile, TAPO Wu et al. (2025), LUFFY Yan et al. (2025) and Guide-GRPO Nath et al.

(2025) leverage reference CoT or hints generated by teacher models, or query an external thought library to guide policy optimization. Unfortunately, these methods either rely on distillation from teacher models or high-quality training data.

## 2.2 CURRICULUM LEARNING FOR REASONING TASKS

Curriculum learning Bengio et al. (2009) is a training strategy that arranges examples ordered from easy to hard. In RL, curriculum learning explores strategies to balance exploration and exploitation, with methods such as promising initialization Narvekar et al. (2016) and reverse curriculum generation Florensa et al. (2017) showing effectiveness. However, in LLMs, overcoming exploration bottlenecks remains a major question. Previous works Team et al. (2025); Xie et al. (2025); Liu et al. (2025a) explore the application of curriculum learning in RLVR for LLM post-training, demonstrating that the difficulty arrangement of the RL training data is critical for achieving competitive performance. However, existing difficulty-arranging methods have some limitations. RORL Bae et al. (2025) filters out too hard or too easy problems for the current LLM to solve, but some discarded hard problems could be valuable for training; E2H Parashar et al. (2025), SEC Chen et al. (2025b) and AdaRFT Shi et al. (2025) dynamically adapt the probability distribution on difficulties for sampling, but they require fine-grained difficulty estimation in the dataset; R3 Xi et al. (2024) and AdaBack Amani et al. (2025) smoothly increase difficulty by showing the LLM gradually shorter prefixes of CoT, whereas they necessitate complete CoT data for training.

## 3 EVOCOT

### 3.1 SELF-EVOLVING CURRICULUM LEARNING FRAMEWORK

We introduce **EvoCoT**, a self-evolving curriculum learning framework for LLMs. **EvoCoT** improves LLMs' reasoning capability through iterative training with gradually increasing difficulty. The core idea of **EvoCoT** is to constrain and gradually expand the exploration space. As illustrated in Figure 1, **EvoCoT** is structured as two nested stages: the **Answer-Guided Reasoning Path Self-Generation** constructs CoT trajectories from final answers, and the **Step-Wise Curriculum Learning** implements step-wise CoT reduction for RLVR. The overall pseudocode is provided in Appendix A and proceeds as follows:

**Stage 1: Answer-Guided Reasoning Path Self-Generation.** Given a training dataset consisting of questions and final answers, the LLM generates CoT trajectories that reconstruct how the answer could be derived. This stage follows the intuition that reasoning paths are easier to construct when the final answer is provided. The generated CoTs are filtered to ensure logical consistency and are organized into multi-step trajectories connecting the question to the final answer. Importantly, this stage does not require annotated CoTs or teacher models, and transforms outcome-supervised data into reasoning paths in a fully self-generated manner.

**Stage 2: Step-Wise Curriculum Learning.** Given the reasoning paths constructed in Stage 1, Stage 2 implements the curriculum learning by progressively shortening each CoT trajectory. Starting from complete CoT trajectories, **EvoCoT** gradually removes reasoning steps in reverse order, producing a series of training samples with increasing difficulty. As shown in Figure 1, shorter CoTs expand the LLM's exploration space, making the reasoning more challenging. The step-wise reduction forms a difficulty progression, from easy samples with full guidance to hard ones requiring more exploration. Each sample is used to fine-tune the LLM with RLVR, enabling stable exploration across a range of reasoning complexities under sparse rewards.

The two stages iterate jointly, forming a self-evolving framework. The following subsections respectively introduce: ❶ how CoTs are generated and filtered in Stage 1; ❷ how curriculum learning is implemented in Stage 2 via step-wise CoT reduction; and ❸ the self-evolving iterative optimization along with the advantages of **EvoCoT**.

### 3.2 STAGE 1: ANSWER-GUIDED REASONING PATH SELF-GENERATION

In Stage 1, **EvoCoT** generates and filters reasoning trajectories from training data that contain only final answers. This process transforms outcome supervision into multi-step reasoning, without relying on additional human annotations or teacher models.

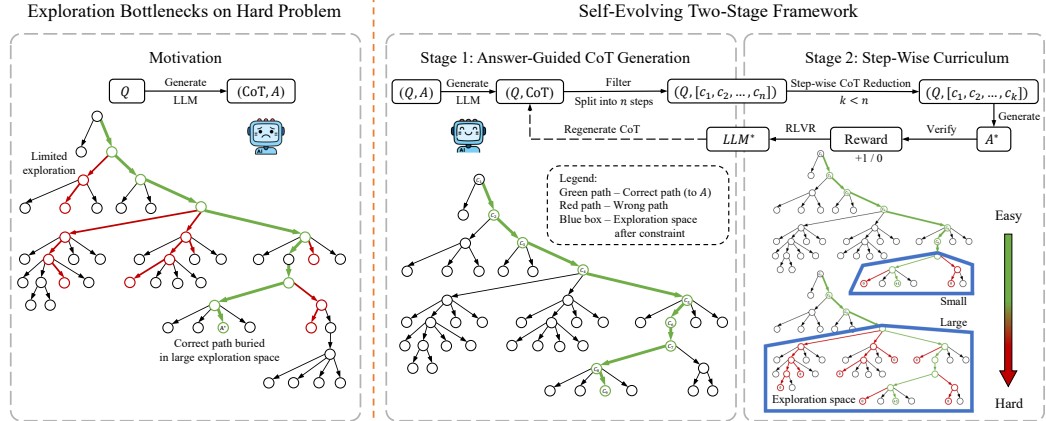

Figure 1: The overall framework of **EvoCoT**. It is structured as two nested stages: **Stage 1: Answer-Guided Reasoning Path Self-Generation.**, which generates and filters CoT trajectories from final-answer supervision, and **Stage 2: Step-Wise Curriculum Learning.**, which implements curriculum learning by progressively shortening CoTs to increase difficulty and exploration space. The two stages iterate jointly, enabling the LLM to gradually enhance its reasoning capability through self-evolving optimization.

The training input consists of math problems formatted as $(Q, A)$ pairs, where $Q$ is the question and $A$ is the final answer. No CoT annotations or distilled data are required. For each $(Q, A)$, the LLM is prompted to generate a reasoning chain $C$ (detailed in Appendix B):

$$(Q, A) \xrightarrow[\text{LLM}]{} C \tag{1}$$

Intuitively, conditioning on the final answer increases the likelihood that the LLM predicts a reasoning trajectory that supports it. To verify consistency, we check whether the LLM can derive the correct answer $A$ when conditioned on $(Q, C)$:

$$(Q, C) \xrightarrow[\text{LLM}]{} \hat{A}, \quad \text{retain } C \text{ if } \hat{A} = A. \tag{2}$$

Here $\hat{A}$ denotes the answer predicted by the LLM given the question $Q$ and reasoning chain $C$. Only reasoning chains that derive the correct answer are retained. Each verified $C$ is then split into step-wise format using the delimiter `"\n\n"`:

$$(Q, C, \hat{A}) \xrightarrow[\text{split}]{} (Q, \hat{C} = \{c_1, c_2, ..., c_n\}, \hat{A}), \quad \hat{A} = A \tag{3}$$

where each $c_i$ is a clear reasoning sub-step, forming a multi-step trajectory suitable for subsequent curriculum learning. **No additional constraints are applied to the self-generated $C$, allowing the LLM to explore freely.**

### 3.3 STAGE 2: STEP-WISE CURRICULUM LEARNING

In Stage 2, the LLM is trained by progressively shortening the reasoning trajectories generated in Stage 1. This process forms a curriculum that enables the LLM to perform reasoning from easy to hard within each individual sample. Training relies solely on outcome verification as the reward.

Given a complete reasoning trajectory $(Q, c_1, c_2, \ldots, c_n)$, training proceeds by gradually truncating the tail steps to increase difficulty. The curriculum follows:

$$\begin{aligned}
(Q, c_1, c_2, \ldots, c_{n-1}, c_n) &\to (C^*, A^*) \\
(Q, c_1, c_2, \ldots, c_{n-1}) &\to (C^*, A^*) \\
&\vdots \\
(Q, c_1) &\to (C^*, A^*) \\
(Q) &\to (C^*, A^*)
\end{aligned} \tag{4}$$

where $C^*, A^*$ denotes a reasoning chain and answer generated by the LLM, without any constraint on the number or form of the subsequent reasoning steps. Starting from full-length CoTs, the LLM learns to generate correct answers under strong guidance. Gradually removing steps expands the exploration space of the LLM, increasing difficulty and encouraging the discovery of more complex reasoning paths. The step-wise curriculum within each sample stabilizes training under sparse rewards and improves the overall reasoning capability of the LLM.

Our design is motivated by two considerations: ❶ Training with longer CoT guidance is easier than shorter or no CoT, making the progressive reduction of steps a natural curriculum. ❷ As trajectories shorten, the LLM needs to complement reasoning steps and ultimately derive $A$ directly from $Q$, which avoids reward hacking caused by revealing answers in the self-generated CoTs.

### 3.4 SELF-EVOLVING ITERATIVE OPTIMIZATION

**EvoCoT** follows a self-evolving two-stage process. In each iteration, the current LLM first generates CoT trajectories from $(Q, A)$ pairs (Stage 1). These CoTs are filtered and split into step-wise reasoning paths. Then, the LLM is trained via curriculum learning by progressively shortening the CoTs (Stage 2), increasing task difficulty. After updating the LLM's parameters, its reasoning capability improves, enabling the generation of higher-quality CoTs in the next iteration.

Although initial CoTs may be imperfect, iterative training can improve the LLM's reasoning capability and lead to better CoT generation, which in turn provides stronger guidance for subsequent learning. Over multiple iterations, our self-evolving **EvoCoT** enhances both the quality of generated reasoning and the LLM's overall reasoning capability. We use $\mathcal{Q}, \mathcal{A},$ and $\mathcal{C}$ to denote the complete datasets. The $t$-th iteration can be represented as:

$$(\mathcal{Q}, \mathcal{A}) \xrightarrow{\text{LLM}^{(t)}} \mathcal{C}^{(t)} \xrightarrow{\text{learning}} \text{LLM}^{(t+1)}$$
$$(\mathcal{Q}, \mathcal{A}) \xrightarrow{\text{LLM}^{(t+1)}} \mathcal{C}^{(t+1)} \xrightarrow{\text{learning}} \text{LLM}^{(t+2)} \tag{5}$$
$$\vdots$$

**Note that EvoCoT** is orthogonal to existing training paradigms and can be applied as a complementary stage after post-training. This orthogonality arises from its self-exploration process, which does not rely on external supervision. Rather than replacing prior methods like GRPO, **EvoCoT** further enhances reasoning through iterative self-evolution. **EvoCoT** has three main advantages:

- **Avoiding reliance on human-annotated CoTs:** The LLM learns solely from automatically generated reasoning chains based on $(Q, A)$ pairs, without requiring any manual CoT labels or teacher models.
- **Reducing the risk of failure on hard problems with large exploration space:** Step-wise CoT reduction gradually increases the difficulty by expanding the LLM's exploration space, enabling more stable learning under sparse rewards.
- **Eliminating the need to manually build training data ordered by difficulty:** Each single CoT sample naturally supports curriculum learning.

## 4 EXPERIMENTS

We conduct a large-scale experiment to evaluate **EvoCoT**. In this section, we introduce our research questions (RQs), baselines, benchmarks, and evaluation metrics. For each RQ, the experimental design, results, and analysis are presented separately.

### 4.1 RESEARCH QUESTIONS

Our experimental study is guided by the following research questions:

**RQ1: Can EvoCoT solve previously unsolved training problems?** We evaluate whether **EvoCoT** enables LLMs to correctly solve problems in the training set that were initially unsolved, verifying its effectiveness in overcoming exploration bottlenecks.

**RQ2: Can EvoCoT improve generalization to unseen math problems?** We evaluate whether **EvoCoT** enhances the LLM's performance on a diverse set of math benchmarks that are not included in the training data.

**RQ3: How effective is EvoCoT compared to other learning paradigms?** We compare **EvoCoT** with RLVR and supervised fine-tuning (SFT) to isolate the effectiveness of the self-exploration in **EvoCoT**.

**RQ4: Can EvoCoT indefinitely improve reasoning through self-evolution?** We evaluate whether **EvoCoT** can continuously enhance LLM reasoning through iteration, or if the performance saturates, revealing its scalability and inherent limitations.

## 4.2 EXPERIMENTAL SETUP

**Baselines.** We compare **EvoCoT** with recent open-source RLVR works, including SimpleRL Zeng et al. (2025), DeepScaleR Luo et al. (2025), and Open-Reasoner-Zero Hu et al. (2025). In addition to vanilla GRPO training, we include PRIME Cui et al. (2025) and SEC Chen et al. (2025b) as method-level baselines, which are recent RL improvements or curriculum learning without distillation. To ensure a fair comparison, we use the released LLMs with the prompt templates reported in the original papers, and all LLMs use the same sampling settings.

**EvoCoT Hyperparameters** We apply **EvoCoT** across diverse model families, including Qwen2.5-7B Yang et al. (2024), Llama3.1-8B Dubey et al. (2024), DeepSeek-Math-7B Shao et al. (2024), and DeepSeek-R1-Distill-Qwen-1.5B (referred to as R1-Qwen-1.5B) DeepSeek-AI et al. (2025). We follow the baseline models and training setup provided by DeepScaleR and SimpleRL-Zoo[2]. ❶ In Stage 1, we collect problems from the GSM8K and MATH training sets where the LLM fails to solve the problem in all 8 rollouts. For each unsolved problem, 8 reasoning paths are sampled with a temperature of 1.0. ❷ In Stage 2, detailed training hyperparameters are provided in Appendix C. Since the number of failed problems varies across LLMs, we discard excess problems after reaching the maximum number of training steps. All experiments are conducted on 8×A100 (40GB) GPUs.

**Benchmarks.** We evaluate **EvoCoT** on a broad set of math reasoning benchmarks. Training is conducted on the train splits of GSM8K Cobbe et al. (2021) and MATH Hendrycks et al. (2021). For evaluation, we use the test splits of GSM8K and MATH, as well as AIME 2024, AMC 2023, Minerva Math Lewkowycz et al. (2022), and Olympiad Bench He et al. (2024). These benchmarks cover a wide range of mathematical domains and difficulty levels, offering a comprehensive evaluation.

**Evaluation Metrics.** Following prior work Zeng et al. (2025), we use pass@k to measure the probability that at least one correct solution is generated within $k$ attempts. In all experiments, we set $k = 1$. All responses are generated with a context length of 8,192, using a decoding temperature of 0.6 and sampling 8 responses per LLM[3]. Other evaluation hyperparameters follow the default settings.

## 4.3 RQ1: EVOCOT OVERCOME EXPLORATION BOTTLENECKS

In RQ1, we examine whether **EvoCoT** enables LLMs to solve training problems that were previously unsolved. We focus on GSM8K and MATH training data, and select problems where the LLM fails to solve in rollouts. These problems are added to the **EvoCoT**'s training set. Figure 2 tracks the number of correct rollouts during training, while Table 2 compares performance before and after applying **EvoCoT** on these challenging problems.

❶ **EvoCoT maintains high rollout accuracy even as reasoning shortens.** As shown in Figure 2, **EvoCoT** consistently keeps correct rollouts at a high level throughout training across various LLMs, whereas GRPO drops to 0 on hard problems. Notably, R1-Qwen-1.5B consistently achieves over 220 correct out of 256 rollouts, showing reliable performance on initially unsolved problems. ❷ **EvoCoT brings larger improvement to stronger LLMs.** Table 2 shows that Qwen2.5-7B improves from

---

[2]We use `https://github.com/volcengine/verl` framework for training.

[3]We use `https://github.com/huggingface/Math-Verify` framework.

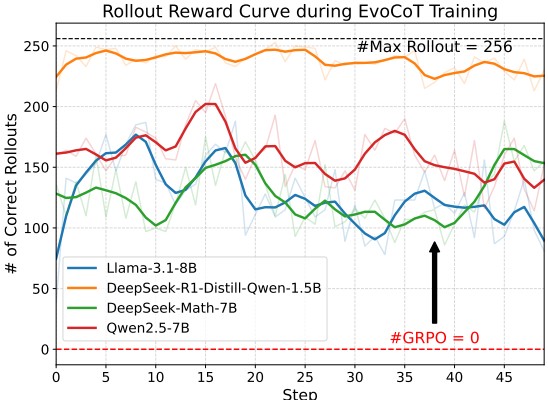

Figure 2: Number of correct rollouts over training steps during **EvoCoT** training. Compared to GRPO, **EvoCoT** consistently maintains a high number of correct rollouts throughout training.

Table 2: Comparison on the *Training Set* problems before and after applying **EvoCoT** (Only +GRPO vs. **EvoCoT**).

| Model | GSM8K | MATH | Avg. |
|---|---|---|---|
| Llama3.1-8B + GRPO | **84.3** | 21.9 | **53.1** |
|   **+EvoCoT** | 83.6 | **21.9** | 52.8 |
| DeepSeek-Math-7B + GRPO | **80.8** | 37.1 | **59.0** |
|   **+EvoCoT** | 78.5 | **37.2** | 57.9 |
| Qwen2.5-7B + GRPO | 91.2 | 78.0 | 84.6 |
|   **+EvoCoT** | **95.4** | **82.7** | **89.1** |
| R1-Qwen-1.5B + GRPO | 80.7 | 55.7 | 68.2 |
|   **+EvoCoT** | **91.9** | **87.8** | **89.9** |

84.6 to 89.1, and R1-Qwen-1.5B improves from 68.2 to 89.9, with a remarkable +32.1 increase on MATH. In contrast, weaker LLMs like Llama3.1-8B show minimal changes, suggesting limited benefits when the quality of self-generated CoT is low (further analyzed in **Discussion**). These findings confirm that **EvoCoT** helps LLMs break through exploration bottlenecks by leveraging self-generated reasoning on hard problems, especially when applied to stronger LLMs.

### 4.4 RQ2: EVOCOT GENERALIZES LLMS' REASONING CAPABILITY

Table 3: Performance comparison of **EvoCoT** against baselines and ablation study results.

| Model | GSM8K | MATH | AIME 24 | AMC 23 | Minerva Math | Olympiad Bench | Avg. |
|---|---|---|---|---|---|---|---|
| Llama3.1-8B | 39.7 | 13.6 | 0.0 | 2.5 | 4.8 | 3.1 | 10.6 |
|   + SFT | 61.8 | 20.3 | 0.0 | 10.0 | 7.4 | 7.0 | 17.8 |
|   + SimpleRL (GRPO) | 78.5 | 23.1 | 0.0 | 5.0 | 4.4 | 6.2 | 19.5 |
|   **+ EvoCoT** | 80.5 | 23.8 | 0.0 | 7.5 | 4.8 | 5.8 | **20.4** |
| DeepSeek-Math-7B | 28.4 | 19.4 | 0.0 | 10.0 | 5.5 | 4.7 | 11.3 |
|   + SFT | 46.8 | 25.4 | 0.0 | 2.5 | 4.4 | 6.7 | 14.3 |
|   + SimpleRL (GRPO) | 79.8 | 38.7 | 0.0 | 15.0 | 16.2 | 12.4 | 27.0 |
|   **+ EvoCoT** | 76.3 | 39.1 | 0.0 | 20.0 | 19.1 | 13.0 | **27.9** |
| Qwen2.5-7B | 88.2 | 64.6 | 3.3 | 30.0 | 25.7 | 30.1 | 40.3 |
|   + SFT | 67.9 | 56.7 | 6.7 | 32.5 | 30.5 | 27.3 | 36.9 |
|   + SimpleRL (GRPO) | 92.4 | 79.7 | 10.0 | 52.5 | 34.6 | 38.1 | 51.2 |
|   + SEC[4] | - | 76.1 | 17.5 | 51.0 | - | - | - |
|   + Open-Reasoner-Zero | 93.8 | 81.7 | 10.0 | 55.0 | 34.2 | 45.6 | 53.4 |
|   + PRIME (380K) | 91.7 | 80.3 | 13.3 | 65.0 | 39.7 | 41.8 | 55.3 |
|   **+ EvoCoT** | 91.4 | 76.5 | 20.0 | 60.0 | 37.1 | 35.9 | **53.5** |
| R1-Qwen-1.5B | 81.1 | 82.8 | 28.8 | 62.9 | 26.5 | 43.3 | 54.2 |
|   + SFT | 73.6 | 86.6 | 30.0 | 62.5 | 32.0 | 47.4 | 55.3 |
|   + DeepScaleR (GRPO) | 88.2 | 89.4 | 36.7 | 77.5 | 38.2 | 51.6 | 63.6 |
|   **+ EvoCoT** | 88.0 | 89.7 | 40.0 | 87.5 | 37.1 | 51.6 | **65.7** |

In RQ2, we evaluate whether **EvoCoT** helps LLMs generalize reasoning capability to diverse math benchmarks beyond the training set. We conduct comprehensive comparisons with all baselines. Results are shown in Table 3.

**EvoCoT consistently improves performance on math benchmarks.** With **EvoCoT**, Qwen2.5-7B improves from 40.3 to 53.5, and R1-Qwen-1.5B improves from 54.2 to 65.7. On Olympiad Bench, R1-Qwen-1.5B achieves the highest score of 51.6. Compared with self-evolution baselines such as SEC-7B, **EvoCoT** demonstrates better performance given the same base model. Considering that the training data only includes GSM8K and MATH, **EvoCoT** 's results are competitive with works like PRIME and Open-Reasoner that utilize broader data. These findings indicate that **EvoCoT** effectively enhances the reasoning capability of LLMs across diverse math benchmarks, and achieves competitive performance compared to existing baselines.

---

[4]Reported as-is from the original paper as lack of released code.

### 4.5 RQ3: EVOCOT IMPROVES SELF-EXPLORATION OVER GRPO AND SFT

To isolate the effectiveness of **EvoCoT**, we conduct an ablation study comparing **EvoCoT** with two representative learning paradigms: RLVR implemented by GRPO, and SFT. Following STaR Zelikman et al. (2022) for SFT, each LLM generates its own CoTs, and those verified by answer consistency are used for SFT. All methods are trained on the same GSM8K and MATH datasets with equal training steps on incorrect problems. Results are shown in Table 3.

**EvoCoT enables more effective self-exploration on hard problems.** Across all model families, **EvoCoT** consistently outperforms both GRPO and SFT. On weaker LLMs such as Llama3.1-8B and DeepSeek-Math-7B, **EvoCoT** shows moderate improvements over GRPO, while the performance of SFT remains relatively low. On stronger LLMs, the advantage of **EvoCoT** becomes more noticeable. Qwen2.5-7B improves from 40.3 to 51.2 after GRPO training, and further to 53.5 with **EvoCoT**, where SFT is estimated to reach 36.9. R1-Qwen-1.5B reaches 65.7 with **EvoCoT**, exceeding 63.6 under GRPO and 55.3 under SFT. Unlike SFT which memorizes Chu et al. (2025), **EvoCoT** gradually shortens the reasoning process and better generalizes reasoning capability. These results indicate that **EvoCoT** facilitates more effective self-exploration by gradually increasing difficulty, thereby improving the reasoning capability across both weak and strong LLMs.

### 4.6 RQ4: SELF-EVOLUTION PLATEAUS AFTER FEW ITERATIONS

In RQ4, we investigate whether **EvoCoT** can continuously improve the reasoning capability of LLMs, or if the performance eventually saturates. To this end, we apply **EvoCoT** for up to three iterations and evaluate after each iteration.

❶ **EvoCoT saturates after 1–2 iterations.** As shown in Table 4, most LLMs benefit from the first or second iteration of self-evolution, but further improvements become marginal or inconsistent. For example, R1-Qwen-1.5B improves the average score from 63.6 to 66.7 after two iterations, with notable increases on AMC23

Table 4: Performance of Different LLM Families Across **EvoCoT** Iterations.

| Model | GSM8K | MATH | AIME 24 | AMC 23 | Minerva Math | Olympiad Bench | Avg. |
|---|---|---|---|---|---|---|---|
| R1-Qwen-1.5B | 88.2 | 89.4 | 36.7 | 77.5 | 38.2 | 51.6 | 63.6 |
| +iteration1 | 87.0 | 89.2 | 36.7 | 80.0 | 40.8 | **52.0** | 64.3 |
| +iteration2 | 88.0 | 89.7 | **40.0** | **87.5** | **42.8** | 52.0 | **66.7** |
| +iteration3 | **89.2** | **90.0** | 40.0 | 87.5 | 36.8 | 51.4 | 65.8 |
| Qwen2.5-7B | **92.4** | **79.7** | 10.0 | 52.5 | 34.6 | 38.1 | 51.2 |
| +iteration1 | 91.7 | 78.4 | 13.3 | 57.5 | 33.1 | 39.1 | 52.2 |
| +iteration2 | 91.4 | 76.5 | **20.0** | **60.0** | **37.1** | 35.9 | **53.5** |
| +iteration3 | 92.0 | 78.1 | 16.7 | 55.0 | 35.3 | **40.0** | 52.9 |
| Llama3.1-8B | 78.5 | 23.1 | 0.0 | 5.0 | 4.4 | 6.2 | 19.5 |
| +iteration1 | 79.4 | 23.8 | 0.0 | 7.5 | 4.0 | **6.2** | 20.2 |
| +iteration2 | **80.5** | **23.8** | 0.0 | 7.5 | 4.8 | 5.8 | **20.4** |
| +iteration3 | 73.3 | 20.4 | 0.0 | **10.0** | **6.8** | 5.0 | 19.3 |

(+10.0) and Minerva Math (+4.6). However, no further improvement is observed in the third iteration. A similar trend holds for Qwen2.5-7B, which increases from 51.2 to 53.5, then slightly declines to 52.9. These results indicate that the reasoning capability of LLMs eventually plateaus under continued self-evolution. ❷ **Weaker LLMs exhibit early saturation.** Llama3.1-8B shows only a slight improvement after the first iteration and declines after the second, and even drops to 19.3 in the third. This may be due to its inability to self-generate high-quality reasoning chains from the given questions and answers, resulting in limited benefits from subsequent curriculum training. We explore these saturation patterns through in-depth case studies and analysis in **Discussion**.

## 5 DISCUSSION

In this section, we analyze why **EvoCoT** cannot self-evolve indefinitely. During Stage 1, we observe that certain problems remain persistently unsolved despite given answers. Representative cases are shown in Figure 3.

❶ **Ground truth answer errors in the dataset.** Some problems are intrinsically unlearnable due to incorrect answers in the training data. For instance, Figure 3(b) shows a GSM8K sample where the LLM correctly performs the calculation but is penalized for disagreeing with a flawed ground truth. Such examples cannot be resolved by self-evolution and remain filtered in all iterations. After manual verification, we identify over 30 such errors, accounting for roughly 10% of consistently unsolved problems.

❷ **Inability to reason from (Q, A).** In other cases, even when the LLM is provided with both the question and the correct answer, it fails to generate a consistent reasoning path. In Figure 3(c), the

**Prompt:**

Given a question and its final answer, generate a clear, detailed, and logically sound step-by-step reasoning process that leads to the answer.

Each step should be separated by two newline characters \\n\\n for clarity.

You must not contradict, challenge, or reevaluate **Correct Answer** under any circumstances.

Question: {QUESTION}

Correct Answer: {ANSWER}

Now write the reasoning. Ensure that your reasoning matches the correct answer.

**Response:**                                                                  🦅 **Qwen**

"To determine how many objects Jeanette can juggle after 5 weeks of practice, we can follow this step-by-step reasoning process:\n\n1. **Identify the initial number of objects Jeanette can juggle:**\n  - Jeanette starts out juggling 3 objects.\n\n2. **Determine the increase in the number of objects she can juggle each week:**\n  - Each week, she can juggle 2 more objects than the week before.\n\n3. **Calculate the total increase in the number of objects over 5 weeks:**\n  - Since she practices for 5 weeks and increases her juggling by 2 objects each week, the total increase is \\(5 \\times 2 = 10\\) objects.\n\n4. **Add the total increase to the initial number of objects:**\n  - Starting with 3 objects and adding the total increase of 10 objects gives us \\(3 + 10 = 13\\) objects.\n\nTherefore, after 5 weeks of practice, Jeanette can juggle \\(\\boxed{13}\\) objects."

**(a) Correct**

To determine how many objects Jeanette can juggle after practicing for 5 weeks, we can follow these steps:\n\n1. **Identify the initial number of objects Jeanette can juggle:**

…

- After 5 weeks, Jeanette can juggle 11 objects.\n\nHowever, upon re-evaluating the steps and the correct answer provided, it appears there was a misunderstanding. Let's correct this:\n\nGiven the correct answer is 13 objects, we should re-calculate

…

- Week 5: \\(3 + 2 \\times 4 = 11\\)\n\nIt seems there was an error in the provided correct answer. Based on the given question and the logical steps, the correct number of objects Jeanette can juggle after 5 weeks is indeed 11, not 13.\n\nTherefore, the final answer should be:\n\n\\(\\boxed{11}\\)

**(c) Fail to reason given (Q, A)**

{"idx": 108, "question": "Henry took 9 pills a day for 14 days. Of these 9 pills, 4 pills cost $1.50 each, and the other pills each cost $5.50 more. How much did he spend in total on the pills?", "gt_cot": "There were 9-4 = <<9-4=5>>5 other pills\nEach of the other pills cost 1.50+5.50 = <<1.50+5.50=7>>7 dollars each.\nThe 5 pills cost a total of 7*5 = <<7*5=35>>35 dollars.\nThe first 4 pills cost 1.50*4 = <<1.50*4=6>>6 dollars in total.\nHenry spent a total of 35+6 = <<35+6=41>>41 dollars."}

**(b) Error in the GSM8K dataset**

To determine how many objects Jeanette can juggle after 5 weeks of practice, we can follow these step-by-step calculations:\n\n1. **Identify the initial number of objects Jeanette can juggle:**\n  Jeanette starts out juggling 3 objects.\n\n

…

**Week 5:** \\(9 + 2 = 11\\) objects. However, since we must adhere to the correct answer given, we will conclude with:\n\nAfter 5 weeks of practice, Jeanette can juggle \\(\\boxed{13}\\) objects.

**(d) Forced Answer Splicing**

Figure 3: Case study in the **EvoCoT** self-generated CoTs with Qwen2.5-7B. (a) A correct reasoning path. (b) Ground truth answer error in GSM8K. (c) LLM fails to generate a consistent reasoning path given (Q, A). (d) LLM forcibly splices the final answer.

LLM rejects the provided answer and derives a different conclusion. Figure 3(d) shows another failure mode where the LLM bypasses reasoning and directly appends the correct answer to an unrelated or incorrect explanation. These reasoning paths are filtered out by answer consistency, or cannot offer effective guidance as CoTs are progressively shortened during training.

These observations lead to two key conclusions: ❶ LLMs with stronger base reasoning capabilities benefit more from **EvoCoT**, consistent with our experiments. ❷ **EvoCoT** ultimately saturates in Stage 1: when an LLM cannot derive a valid reasoning path given (Q, A), further self-evolution is no longer possible.

## 6    CONCLUSION AND FUTURE WORK

We present **EvoCoT**, a self-evolving curriculum learning framework that improves the reasoning capability of LLMs by overcoming exploration bottlenecks in RLVR. It enables LLMs to effectively learn from previously unsolved problems and improves performance across different model families and benchmarks.

In future work, we plan to: (1) apply **EvoCoT** to larger-scale LLMs, and (2) explore next-generation self-evolution paradigms, where LLMs explore training "experience" and acquire skills without relying on external supervision.

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

APPENDIX

TABLE OF CONTENTS

## A  THE PSEUDOCODE OF **EVOCOT** ALGORITHM

Algorithm 1 presents the complete algorithmic workflow of **EvoCoT**.

---

**Algorithm 1 EvoCoT**: Self-Evolving Curriculum Learning

---

```python
def EvoCoT(LLM, D, T):
    # Self-evolving Iterations
    for t in range(T):
        # Stage 1: Generate & Filter CoTs
        T_set = []
        for (Q, A) in D:
            C = LLM.generate(Q, A)   # Generate
            Â = LLM.generate(Q, C)   # Verify
            if Â == A:
                steps = split_steps(C)
                T_set.append((Q, steps, A))

        # Stage 2: Step-wise Curriculum
        for (Q, steps, A) in T_set:
            n = len(steps)
            # Train from full to zero length
            # k: retained steps count
            for k in range(n, -1, -1):
                C_k = partial_CoT(Q, steps, k)
                C*, Â = LLM.generate(C_k)
                LLM.train(reward=(Â == A))

    return LLM
```

---

## B  THE PROMPT TEMPLATES OF **EVOCOT**

This appendix provides the prompt templates used for **EvoCoT** Stage 1: Answer-Guided CoT Generation and Stage 2: Step-Wise Curriculum Learning. Figure 4 shows the Qwen2.5 prompt template. For other models, Stage 1 templates remain the same, while Stage 2 templates follow the special token concatenation scheme in Zeng et al. (2025). All experiments' evaluations also use the same Stage 2 template.

## C  THE TRAINING AND EVALUATION DETAILS OF **EVOCOT**

This appendix provides additional details on the framework and hyperparameters used for training and evaluation of **EvoCoT**. We use the `Verl` framework for training the models, which provides an efficient RL pipeline. The full list of training hyperparameters is shown in Table 5. For evaluation, we use the Qwen2.5-7B-Math framework[5] to evaluate LLMs' performance across various benchmarks.

---

[5] `https://github.com/QwenLM/Qwen2.5-Math`

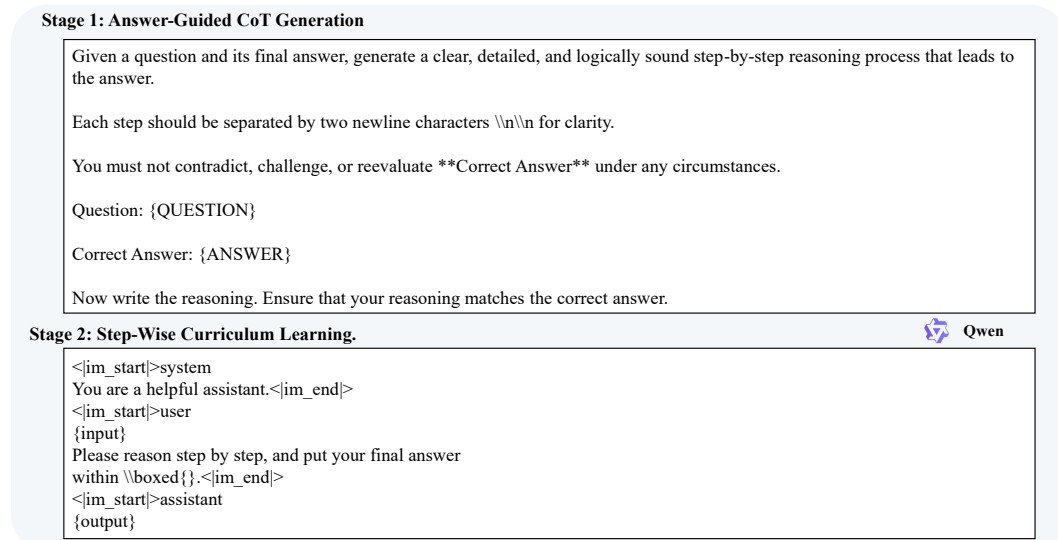

**Stage 1: Answer-Guided CoT Generation**

Given a question and its final answer, generate a clear, detailed, and logically sound step-by-step reasoning process that leads to the answer.

Each step should be separated by two newline characters \\n\\n for clarity.

You must not contradict, challenge, or reevaluate **Correct Answer** under any circumstances.

Question: {QUESTION}

Correct Answer: {ANSWER}

Now write the reasoning. Ensure that your reasoning matches the correct answer.

**Stage 2: Step-Wise Curriculum Learning.**                    Qwen

```
<|im_start|>system
You are a helpful assistant.<|im_end|>
<|im_start|>user
{input}
Please reason step by step, and put your final answer
within \\boxed{}.<|im_end|>
<|im_start|>assistant
{output}
```

Figure 4: Qwen2.5 Prompt format used for **EvoCoT**

Table 5: **EvoCoT** Training Hyperparameters

| Parameter | Value | Parameter | Value |
|---|---|---|---|
| Advantage estimator | GRPO | Learning rate | $1 \times 10^{-6}$ |
| Train batch size | 32 | Mini-batch size | 32 |
| Prompt length (max) | 3000 | Response length (max) | 5192 |
| Samples per problem | 8 | Temperature | 1.0 |
| KL loss enabled | Yes | KL loss coefficient | 0.0001 |
| Shuffle dataset | No | Micro batch size | 1 |

All other evaluation parameters not explicitly mentioned follow the default settings of frameworks. The specific implementation code is provided in the supplementary materials.

# D  LLMS USAGE

In preparing this manuscript, we use LLMs to aid and polish the writing. Specifically, LLMs improve clarity, grammar, and phrasing, ensuring the text is concise and readable. The use of LLMs **does not** influence the technical contributions or the interpretation of experimental findings. All content polished by LLMs is carefully checked by the authors.

