# OpenReview forum: "EvoCoT: Overcoming the Exploration Bottleneck in Reinforcement Learning for LLMs"
_ICLR.cc/2026/Conference — ICLR 2026 Conference Withdrawn Submission_

### Official Review · Reviewer_gChj · 2025-10-30

**Soundness:** 2
**Presentation:** 3
**Contribution:** 2
**Rating:** 4
**Confidence:** 3

**Summary:**

This paper proposes the EvoCoT framework, which overcomes the exploration bottleneck in reinforcement learning with RLVR for LLMs through a two-stage self-evolving curriculum learning mechanism. The method improves reasoning ability effectively without external supervision and demonstrates its effectiveness across various models and benchmark tasks.

**Strengths:**

- EvoCoT enables stable learning under sparse reward conditions, effectively alleviating the reward sparsity issue commonly seen in RLVR tasks.
- By progressively truncating self-generated CoTs, EvoCoT automatically constructs training samples with gradually increasing difficulty, achieving a natural "easy-to-hard" curriculum. This process eliminates the need for manual difficulty annotation or ordering of training samples, significantly simplifying the training pipeline.

**Weaknesses:**

- Figure 2 in RQ1 may not provide convincing evidence, as EvoCoT is conditioned on partially correct CoTs during rollouts, which naturally leads to more correct outputs. Direct evaluation on a held-out validation set would more effectively demonstrate the method’s impact.
- The generalization experiments in RQ2 are still limited to the mathematical domain. Since reasoning patterns may be highly similar across different math datasets, this setup may not sufficiently demonstrate generalization. Testing on heterogeneous tasks—such as code generation or logical reasoning—would provide more convincing evidence of cross-domain transferability.

**Questions:**

- The two-stage design may significantly increase sampling overhead. In standard GRPO training, one step samples `batch_size × n_sample` trajectories. Does one EvoCoT step consist of the full Stage 1 and Stage 2? Or does it only sample `batch_size` examples from the filtered (Q, C) pairs in Stage 1, then apply `n_sample` rollouts? How does the training time of EvoCoT compare to GRPO when using the same number of training steps? Does EvoCoT still maintain an advantage under equal wall-clock time constraints?
- When the provided CoT is very long, the problem becomes too easy, and the model may achieve perfect accuracy; however, the reward remains sparse. Would this lead to inefficiencies in training?
- Was GRPO trained only on the subset of *hard* problems? If so, the initial gradients may be extremely weak, making the comparison potentially unfair.

---

> ### Author Response · Authors · 2025-11-21
>
> ## **For W1 (Experiment setup)**
>
> In Figure2, the GRPO curve remains at 0 because there are some hard problems in the training set that Qwen2.5-7B still fails to solve (for example, 8.8% of GSM8K and 22.0% of MATH problems). Even with a temperature of 0.6 and 16 rollouts, the model cannot solve these problems. In this case, GRPO remains at 0, which indicates that the training has reached an exploration bottleneck.
>
> In fact, the purpose of the RQ1 experiment is to examine whether EvoCoT can correctly solve the problems that the model fails to solve during rollouts on the training set, and whether it can obtain updated rewards. Compared to GRPO, we do not use more data or additional training steps. EvoCoT enables the model to **make better use of the existing training data** and to overcome sparse reward signals, thereby regaining meaningful gradients. As the most direct validation of EvoCoT’s effectiveness, demonstrating that it overcomes sparse rewards on the training set is exactly the goal of RQ1.
>
> ## **For W2 (Generalization)**
>
> Thank you for the valuable comment. Due to time constraints, we evaluate Qwen2.5-7B with both GRPO and EvoCoT on [LiveCodeBench](https://livecodebench.github.io/) using [Lighteval](https://github.com/huggingface/lighteval). The experimental results are as follows:
>
> | LiveCodeBench | GRPO | EvoCoT |
> |:--:|:-:|:--:|
> | pass@1 |  7.46  |  8.21  |
>
> In future work, we plan to evaluate on additional benchmarks such as [BIG-bench-hard](https://arxiv.org/abs/2210.09261) to further assess cross-domain generalization.
>
> ## **For Q1 (Training Overhead)**
>
> Thank you for raising the concern regarding sampling and training overhead.
>
> We clarify that **Stage-1 of EvoCoT is a one-time preprocessing step**. It is **not** executed repeatedly during RL training. Thus each RL step of EvoCoT does **not** include the cost of generating trajectories from Stage-1 and only uses the verified trajectories prepared in advance.
>
> During RL, each EvoCoT step is identical to **standard GRPO**: sampling `batch_size` items from the `(Q, C)` pool and performing `n_sample` rollouts per item. Therefore, the per-step sampling cost is the same as GRPO. The full procedure is given in **Appendix A, Algorithm 1**.
>
> Under identical hardware and identical numbers of training steps, we observe that EvoCoT increases total training time by only **about 4%** (Qwen2.5-7B, 413 min vs. 427 min), mainly due to filtering rollouts for harder samples. Similar to GRPO, rollouts dominate the computational cost.
>
> Under **the same wall-clock time budget**, EvoCoT still significantly outperforms GRPO, demonstrating that the improvement comes from **more effective exploration and gradient signals**, rather than increased computation.
>
> ## **For Q2 (CoT Length Ablation)**
>
> In fact, in our preliminary experiments, we explicitly examined **how many CoT steps should be removed at each iteration**. As shown in **Appendix A, Algorithm 1, line 18**, we consider:
> `for k in range(n, -1, -1)`. Based on empirical sampling, the initial CoT-length distribution of Qwen2.5-7B is:
> In our preliminary experiments, taking Qwen as an example, the initial CoT-length distribution is:
>
> | Length |  2  |  3  |  4  |  5  |  6  |  7  |  8  |  9  |  10 |  11 |  12 |  13 |  14 |  15 |  16 |  17 |  18 |
> | :--: | :-: | :-: | :-: | :-: | :-: | :-: | :-: | :-: | :-: | :-: | :-: | :-: | :-: | :-: | :-: | :-: | :-: |
> |  Count |  1  |  5  |  4  |  27 |  34 |  40 |  32 |  36 |  33 |  30 |  36 |  28 |  10 |  11 |  11 |  1  |  1  |
>
> All variants adopt the same total number of RL training steps=500. Even if the CoT is not fully shortened, training stops at step 500. The results on Qwen2.5-7B are:
>
> | Step | GSM8K | MATH | AIME | AMC  | Minerva | Olympiad | Avg  |
> | - | - | - | - | - | - | - | - |
> | 1  | 91.4  | 76.5 | 20.0 | 60.0 | 37.1 | 35.9  | 53.5 |
> | 2  | 90.8  | 77.1 | 16.7 | 52.5 | 38.6 | 38.7  | 52.4 |
> | 3  | 91.6  | 77.2 | 13.3 | 55.0 | 31.6 | 36.3  | 50.8 |
>
> A larger shortening step makes each stage substantially harder and disrupts the progression. Therefore, we set the step size to −1.
> This ablation will be added to the revision.
>
> In future work, if applying EvoCoT to stronger flagship models (e.g., DeepSeek-R1), this step size can be dynamically adjusted based on the reward — if the model answers correctly at each step, the step size can be increased, and the number of CoT steps shortened in each iteration can be set to 2, 3, or even more, until the rollouts in a batch achieve a balance between positive and negative samples.
>
> ## **For Q3 (Training Details)**
>
> No, GRPO was also trained on the easier problems, so the initial rewards were not sparse. After one round of training, on the remaining hard problems that the model still cannot solve (for example, 8.8% of GSM8K and 22.0% of MATH problems for Qwen2.5-7B), the phenomenon shown in Figure 1 appears. This is when EvoCoT starts to take effect, and compared to GRPO, EvoCoT avoids sparse rewards.

---

> ### Comment · Reviewer_gChj · 2025-11-26
>
> Thank you for your reply. However, I think you might not have understood my question.
>
> *"When the provided CoT is very long, the problem becomes too easy, and the model may achieve perfect accuracy; however, the reward remains sparse. Would this lead to inefficiencies in training?"*
>
> When the provided correct CoT is too long, the problem becomes quite easy. In this case, the model's rollout is likely to be entirely correct, and there will also be no gradients. How to avoid oversimplifying problems may also be an issue you need to consider in your work.

---

> ### Author Response · Authors · 2025-11-26
>
> Thank you for the insightful follow-up. Your question touches on a key conceptual point.
>
> For Q2, I may not have explained it clearly: **when the provided CoT is long, we also remove more CoT steps at each iteration**. This allows us **to maintain around ~50% rollout accuracy**, instead of letting the model simply solve oversimplifying problems effortlessly with no gradients.
>
> This is the motivation for the ablation on how many CoT steps are removed. For a relatively weaker model such as Qwen2.5-7B, removing one CoT step per iteration is sufficient. In future work, if applying EvoCoT to stronger flagship models (e.g., DeepSeek-R1), the step size can be dynamically adjusted based on the rollout reward.
>
> We will include a more detailed discussion of this point in the revised version. Thank you.

---

### Official Review · Reviewer_XuKm · 2025-11-01

**Soundness:** 3
**Presentation:** 2
**Contribution:** 2
**Rating:** 4
**Confidence:** 4

**Summary:**

This work propose a curriculum learning framework for RLVR reasoning process. In stage 1, LLM generates CoT trajectories conditioned on questions and final answers. The stage 2 progressively shortens the reasoning paths constructed in stage 1.

**Strengths:**

1. The sparse-reward problem in RLVR that this work focuses on is important and worthy of investigation.
2. The proposed curriculum learning method is simple and extensible to prior methods, without the need of human annotations or expert models.

**Weaknesses:**

1. The performance gains over method-level baselines appear modest and, in one case, are lower than PRIME with Qwen2.5-7B (53.5 vs. 55.3).
2. The step-wise procedure in Step 2 seems potentially time-consuming. For a reasoning trajectory of n steps $(Q, c_1, c_2, \ldots, c_n)$, the LLM needs to produce n answers for curriculum learning. How is the value of $n$ determined across different trajectories?

**Questions:**

1. The setup for Figure 2 is a bit unclear. Could you clarify the conditions under which the GRPO curve remains at 0?
2. Could you please report the training time of EvoCoT compared with vanilla GRPO?
3. As illustrated in Figure 1 (Stage 2), shorter CoTs intuitively expand the exploration space. Could you provide an empirical analysis quantifying this exploration improvement? For example, metrics indicating exploration under different CoT lengths would be helpful.

---

> ### Author Response · Authors · 2025-11-21
>
> ## **For W1 (Modest Improvement)**
>
> Thank you for the comment.
>
> First, we clarify that Table 3 compares **final results** trained on **the same base models using different methods**. Many open-source models in the table are trained with **much larger datasets than EvoCoT**. For example, “PRIME (380K)” indicates the size of its training set. In contrast, EvoCoT makes **more effective use of the available data** — it can solve challenging problems in GSM8K and MATH that other mthods fails to address. Considering that our training data **only includes GSM8K and MATH**, EvoCoT remains competitive with models such as PRIME and Open-Reasoner that rely on substantially broader datasets.
>
> Second, as noted in the Introduction, there are few prior works that **(1) do not use additional CoT distillation, (2) do not filter training data, and (3) apply curriculum learning.** Under these constraints, the current state-of-the-art curriculum baseline is SEC, which we adopt. EvoCoT achieves a significant improvement over SEC and other methods in the same category.
>
> The revised version will emphasize these distinctions and will annotate the training data scale for each model to make the comparison clearer.
>
> ## **For W2 and Q2 (Time-consuming)**
>
> Thank you for raising the concern regarding sampling and training overhead.
>
> We clarify that **Stage-1 of EvoCoT is a one-time preprocessing step**. It is **not** executed repeatedly during RL training. Thus each RL step of EvoCoT does **not** include the cost of generating trajectories from Stage-1 and only uses the verified trajectories prepared in advance.
>
> During RL, each EvoCoT step is identical to **standard GRPO**: sampling `batch_size` items from the `(Q, C)` pool and performing `n_sample` rollouts per item. Therefore, the per-step sampling cost is the same as GRPO. The full procedure is given in **Appendix A, Algorithm 1**.
>
> The value of **n** depends on the number of segments in the model-generated reasoning chain. In our preliminary experiments, taking Qwen as an example, the initial CoT-length distribution is:
>
> | Length |  2  |  3  |  4  |  5  |  6  |  7  |  8  |  9  |  10 |  11 |  12 |  13 |  14 |  15 |  16 |  17 |  18 |
> | :----: | :-: | :-: | :-: | :-: | :-: | :-: | :-: | :-: | :-: | :-: | :-: | :-: | :-: | :-: | :-: | :-: | :-: |
> |  Count |  1  |  5  |  4  |  27 |  34 |  40 |  32 |  36 |  33 |  30 |  36 |  28 |  10 |  11 |  11 |  1  |  1  |
>
> Under identical hardware and identical numbers of training steps, we observe that EvoCoT increases total training time by only **about 4%** (Qwen2.5-7B, 413 min vs. 427 min), mainly due to filtering rollouts for harder samples. Similar to GRPO, rollouts dominate the computational cost.
>
> Under **the same wall-clock time budget**, EvoCoT still significantly outperforms GRPO, demonstrating that the improvement comes from **more effective exploration and gradient signals**, rather than increased computation.
>
> We maintain the **same total number of training steps** as GRPO and other baselines. For a single training step, there is **no additional time overhead**: GRPO sees only the question, while EvoCoT rollouts include the model’s own partial CoT. Even if the CoT is not fully shortened, training stops at step 500.
>
> ## **For Q1 (Experiment setup)**
>
> Thank you for the question.
>
> The GRPO curve remains at 0 because there are some hard problems in the training set that Qwen2.5-7B still fails to solve (for example, 8.8% of GSM8K and 22.0% of MATH problems). Even with a temperature of 0.6 and 16 rollouts, the model cannot solve these problems. In this case, GRPO remains at 0, which indicates that the training has reached an exploration bottleneck.
>
> ## **For Q3 (Exploration Space)**
>
> Using CoT length as a curriculum signal has been discussed in prior studies such as **[Coconut](https://arxiv.org/abs/2412.06769)** [1], **[Train Long, Think Short](https://arxiv.org/abs/2508.08940)** [2]. And Google work **[Chain-of-Thought Reasoning Without Prompting](https://arxiv.org/abs/2402.10200)** [3] illustrates how trajectory length correlates with the model’s exploration space, which supports our use of CoT length as curriculum signal.
>
> ---
>
> [1] Shibo Hao, Sainbayar Sukhbaatar, DiJia Su, Xian Li, Zhiting Hu, Jason Weston, Yuandong Tian: Training Large Language Models to Reason in a Continuous Latent Space. CoRR abs/2412.06769 (2024)
>
> [2] Hasan Abed Al Kader Hammoud, Kumail Alhamoud, Abed Hammoud, Elie Bou-Zeid, Marzyeh Ghassemi, Bernard Ghanem: Train Long, Think Short: Curriculum Learning for Efficient Reasoning. CoRR abs/2508.08940 (2025)
>
> [3] Xuezhi Wang, Denny Zhou: Chain-of-Thought Reasoning Without Prompting. NeurIPS 2024

---

> > ### Comment · Reviewer_XuKm · 2025-11-25
> >
> > I thank the authors for their response. Based on the feedback, I still have some questions that need to be answered.
> > 1. The claim about Qwen2.5-7B struggling with 22% of MATH problems seems inconsistent with known performance (should achieve >90% pass@16 on math500 after GRPO). Authors should verify they're using the optimal experimental settings for a fair comparison.
> > 2. The ablation studies comparing method effectiveness and computational costs need more fair and clear comparisons.

---

> > > ### Author Response · Authors · 2025-11-25
> > >
> > > Thank you for the follow-up.
> > >
> > > 1. We appreciate your careful observation. We are using **[the full MATH dataset](https://arxiv.org/pdf/2103.03874)**, not Math500. And our discussion refers specifically to the MATH **training set**, where **the exploration bottleneck occurs in training**. Please see Table 2 in our paper for detail.
> > >
> > > 2. We understand the concern and would like to clarify that all ablations are conducted under strictly controlled settings:
> > >
> > > * Identical hardware
> > >
> > > * Identical total training steps
> > >
> > > * Identical rollout budget
> > >
> > > * No additional filtering or distillation
> > >
> > > While Stage-1 preprocessing introduces extra time, during RL the model sees prompts consisting of ``Q + partial CoT``, and rollouts continue from there. **Because EvoCoT provides partial CoT hints, the rollout length is shorter compared to GRPO starting rollouts directly from Q.** Under such conditions, for a single training step, there is no additional time overhead. If we count the total time including Stage-1, the total training time increases by only about 4%.

---

> > > > ### Comment · Reviewer_XuKm · 2025-11-25
> > > >
> > > > Thank you for your response, and the clarifications have addressed most of my concerns. I have raised my score.

---

> > > > > ### Author Response · Authors · 2025-11-25
> > > > >
> > > > > Thank you very much for raising our score. We really appreciate your support!

---

### Official Review · Reviewer_dwRX · 2025-11-01

**Soundness:** 2
**Presentation:** 3
**Contribution:** 1
**Rating:** 2
**Confidence:** 3

**Summary:**

EvoCoT is a self-evolving framework which utilizes chain of thought trajectories and improve model performance on hard mathematical problems. The first stage extracts logically consistent CoT trajectories and the second stage involves training the model on these trajectories, gradually removing later CoT steps as training progresses. As partial CoT trajectories are harder to learn from than complete ones, this creates a curriculum for more guided learning.

Stage 1: Answer-Guided Reasoning Path Self-Generation: In this stage, the model is shown a prompt and answer and asked to derive the answer. To verify that the generated reasoning chain is correct, the model is then given the prompt and reasoning chain and asked to compute the answer. If the 2 answers don't match, the example is discarded.

Stage 2: Step-Wise Curriculum Learning: We use the chains extracted in the first step to improve the model's reasoning capabilites. To start, the model is shown the complete reasoning trajectory. With every step, one reasoning step is removed in reverse order. As training progresses, the the chains get shorter and the problem gets harder.

The authors demonstrate the effectiveness of EvoCoT on a variety of model families, baselines and benchmarks.

**Strengths:**

- The paper is well-written and easy to understand.
 - It showcases the effectiveness of EvoCoT across a number of model families, benchmarks and baselines.

**Weaknesses:**

EvoCoT has limited novelty. [[1]](https://arxiv.org/pdf/2203.14465) proposes the idea of generating and training on CoT paths with the main difference being that STaR doesn't provide the model with the answer during generation. Similarly, the idea of using truncated versions of the traces to enable harder problem solving has been explored in other works [[2]](https://arxiv.org/abs/2402.05808)[[3]](https://arxiv.org/abs/2506.18110).

The effectiveness of EvoCoT is also not clear. It seems to generalize better to certain tasks over others (AIME 24 and AMC 23 generally improve while others remain the same or regress). It's not obvious why that is the case. The paper would benefit from a deeper analysis into the conditions under which EvoCoT can enable the most gains. One part of this has been explored as we see that a more powerful model generates higher quality trajectories leading to stronger improvements. However, this hypothesis should be tested against larger models as well.

The generation of CoT trajectories and stepwise learning introduces overhead during training. It would be good to include some numbers to quantify this.

EvoCoT seems to suffer from label leakage related training accuracy inflation issues as seen by the strong improvements on the GSM8K and MATH training set in Table 2 but neutral results on the test sets in Table 3.

**Questions:**

Please see weaknesses

---

> ### Author Response · Authors · 2025-11-21
>
> ## **For W1 (Novelty Concern)**
>
> Thank you for the comment. We first clarify that our work studies a **fundamentally different problem**: how to enable the model to explore effectively under sparse reward signals during reinforcement learning. This is intrinsically different from the goals of [[1]](https://arxiv.org/pdf/2203.14465)[[2]](https://arxiv.org/abs/2402.05808)[[3]](https://arxiv.org/abs/2506.18110).
>
> Second, methods such as [[2]](https://arxiv.org/abs/2402.05808)[[3]](https://arxiv.org/abs/2506.18110) assume access to **additional annotation of CoT**, and their learning procedures rely on extra overhead such as **manual annotation** or **teacher model distillation**. These requirements are precisely the limitations discussed in our Introduction, and EvoCoT is explicitly designed to avoid such dependencies.
>
> Third, we also clarify a misunderstanding: **STaR does provide the answer to the model**. In STaR, the model performs CoT on top of its own sampled reasoning traces, which is exactly where our **+SFT baseline** comes from. Thus STaR does not address the sparse-reward exploration problem we focus on.
>
> In summary, our work targets how to enable the model to **self-evolve during exploration** when no additional data is available. EvoCoT evolves its own trajectories without external supervision, making it fundamentally different from prior approaches.
>
> ## **For W2 (Effectiveness)**
>
> Thank you for your attention to the effectiveness of EvoCoT.
>
> First, we believe that the varying improvements across tasks primarily arise from **different levels of dependence on intermediate reasoning**. Harder Tasks such as **AIME24 and AMC23** contain highly multi-step reasoning, allowing EvoCoT’s CoT-based exploration to offer substantial gains. In contrast, tasks where the intermediate steps are less essential or where answer patterns are more fixed (e.g., certain GSM8K subtypes) naturally benefit less, which explains why some results remain unchanged.
>
> Second, across the four model families used in this study, **DeepSeek-R1-Distill-Qwen and the Qwen** are widely recognized as having **stronger reasoning capability**, and therefore show more significant improvements under EvoCoT. Meanwhile, **DeepSeek-Math-7B and Llama3.1-8B** demonstrate comparatively smaller gains. Due to hardware limitations, our experiments currently cover only models at 7B and below. We plan to extend the evaluation to 13B and larger models in future work to systematically validate this hypothesis.
>
> ## **For W3 (Training Overhead)**
>
> Thank you for raising the concern regarding sampling and training overhead.
> We clarify that **Stage-1 of EvoCoT is a one-time preprocessing step**. It is **not** executed repeatedly during RL training. Thus each RL step of EvoCoT does **not** include the cost of generating trajectories from Stage-1 and only uses the verified trajectories prepared in advance.
>
> During RL, each EvoCoT step is identical to **standard GRPO**: sampling `batch_size` items from the `(Q, C)` pool and performing `n_sample` rollouts per item. Therefore, the per-step sampling cost is the same as GRPO. The full procedure is given in **Appendix A, Algorithm 1**.
>
> Under identical hardware and identical numbers of training steps, we observe that EvoCoT increases total training time by only **about 4%** (Qwen2.5-7B, 413 min vs. 427 min), mainly due to filtering rollouts for harder samples. Similar to GRPO, rollouts dominate the computational cost.
>
> Under **the same wall-clock time budget**, EvoCoT still significantly outperforms GRPO, demonstrating that the improvement comes from **more effective exploration and gradient signals**, rather than increased computation.
>
> ## **For W4 (Label Leakage Concern)**
>
> We emphasize that the training data only contains the original problems and the model-generated intermediate reasoning traces. **When we truncate the reasoning chain and train to the end position, the final token the model sees is still the problem**, identical to GRPO. At **no point does the model access the true answer**, so label leakage cannot occur.
>
> The discrepancy between training-set gains and test-set gains mainly reflects **different levels of dependence on intermediate reasoning**. Competition-level tasks such as AIME24 and AMC23 depend heavily on multi-step structured reasoning, and therefore benefit significantly from EvoCoT’s high-quality trajectories. By contrast, some GSM8K subtypes depend less on complex reasoning chains or have more fixed answer patterns, naturally reducing the advantage of EvoCoT and resulting in more neutral test-set outcomes.

---

### Official Review · Reviewer_3jVt · 2025-11-11

**Soundness:** 2
**Presentation:** 2
**Contribution:** 2
**Rating:** 2
**Confidence:** 3

**Summary:**

This paper proposes a new two-stage post-training method to LLMs, where the reasoning chains in training are ordered in a way that is supposedly making the learning from easy to hard.

**Strengths:**

- Significance: curriculum learning is an important direction to study, and current practitioners not sufficiently understand how to design effective curriculum. This paper provides a fresh perspective from CoT lengths.

**Weaknesses:**

- Method: The training method looks to be problematic.
  - First, please provide a clear algorithm on your method illustrating how the LLM is being trained in detail. Example: [[Phuong and Hutter](https://arxiv.org/pdf/2207.09238)]. Eq. 4 and Eq.5 are too toy and hard to understand.
  - Second, the method appears to be **off-policy**? Please provide careful analysis on the potential off-policy effect for the practical algorithm used.
  - Third, the term curriculum learning has traditionally been used to describe the easy-to-hard scheduling/sampling of the marginal distribution of envs/tasks/prompts, e.g., [[Bengio et al., 2009](https://ronan.collobert.com/pub/2009_curriculum_icml.pdf)], [[Parker-Holder et al., 2022](https://arxiv.org/pdf/2203.01302)], [[Ye et al., 2024](https://arxiv.org/pdf/2411.00062)], instead of the conditional distribution as in this work; it would be helpful to add comparisons to help readers to gain a deeper understanding. Also, training loss or reward discrepancies are often used as the proxy for difficulty in prior works, while this work uses the length of reasoning trajectories -- the authors should provide more ablations on this proxy.
- Format: There are some issues with the format. For example, \citep should be used to add parentheses to the citations; the star notation (*) is usually used to denote optimality, and its usage in Eq.4 is confusing.
- Experiments: The experimental results as shown in Table 3 looks to be relatively neutral in general. It would be also helpful to add comparisons to other curriculum-based methods.

References:
[1] Bengio, Yoshua, Jérôme Louradour, Ronan Collobert, and Jason Weston. "Curriculum learning." In Proceedings of the 26th annual international conference on machine learning, pp. 41-48. 2009.
[2] Parker-Holder, Jack, Minqi Jiang, Michael Dennis, Mikayel Samvelyan, Jakob Foerster, Edward Grefenstette, and Tim Rocktäschel. "Evolving curricula with regret-based environment design." In International Conference on Machine Learning, pp. 17473-17498. PMLR, 2022.
[3] Ye, Ziyu, Rishabh Agarwal, Tianqi Liu, Rishabh Joshi, Sarmishta Velury, Quoc V. Le, Qijun Tan, and Yuan Liu. "Scalable Reinforcement Post-Training Beyond Static Human Prompts: Evolving Alignment via Asymmetric Self-Play." arXiv preprint arXiv:2411.00062 (2024).
[4] Pattnaik, Pulkit, Rishabh Maheshwary, Kelechi Ogueji, Vikas Yadav, and Sathwik Tejaswi Madhusudhan. "Curry-dpo: Enhancing alignment using curriculum learning & ranked preferences." arXiv preprint arXiv:2403.07230 (2024).

**Questions:**

Please see the weakness section. I am happy to raise my score if the authors could sufficiently address the concerns raised.

---

> ### Author Response · Authors · 2025-11-21
>
> ## **For W1 (Method – algorithm clarity)**
>
> Thank you for the suggestion. We already provide the full pseudocode in **Appendix A, Algorithm 1**. In the revised version, we will further refine both the pseudocode and the mathematical expressions so that the training loop becomes more precise and easier to follow.
>
> ## **For W2 (Method – off-policy concern)**
>
> EvoCoT does **not** replace the RL sampling distribution with Stage-1 trajectories.
> Stage-1 is only used to **construct and filter candidate CoT trajectories** generated by the behavior policy $\pi_{\text{gen}}$. These trajectories form a prefix pool for Stage-2.
>
> During Stage-2, the current policy continues the reasoning **on top of these prefixes**, and the updates rely solely on the **on-policy continuation part**. The full procedure is shown in **Appendix A, Algorithm 1**.
>
> ## **For W3 (Curriculum learning definition + proxy + ablations)**
>
> Thank you for the valuable comment.
> Using CoT length as a curriculum signal has been discussed in prior studies such as **[Coconut](https://arxiv.org/abs/2412.06769)** [1], **[Train Long, Think Short](https://arxiv.org/abs/2508.08940)** [2]. And Google work **[Chain-of-Thought Reasoning Without Prompting](https://arxiv.org/abs/2402.10200)** [3] illustrates how trajectory length correlates with the model’s exploration space, which supports our use of this proxy.
>
> In the revision, we will clarify the distinction between our conditional-distribution curriculum and the traditional marginal-distribution curriculum, and add more explanation to make the contrast explicit.
>
> ### **Ablation on the length-based proxy**
>
> In fact, in our preliminary experiments, we explicitly examined **how many CoT steps should be removed at each iteration**. As shown in **Appendix A, Algorithm 1, line 18**, we consider:
> `for k in range(n, -1, -1)`. Based on empirical sampling, the initial CoT-length distribution of Qwen2.5-7B is:
>
> | Length |  2  |  3  |  4  |  5  |  6  |  7  |  8  |  9  |  10 |  11 |  12 |  13 |  14 |  15 |  16 |  17 |  18 |
> | :----: | :-: | :-: | :-: | :-: | :-: | :-: | :-: | :-: | :-: | :-: | :-: | :-: | :-: | :-: | :-: | :-: | :-: |
> |  Count |  1  |  5  |  4  |  27 |  34 |  40 |  32 |  36 |  33 |  30 |  36 |  28 |  10 |  11 |  11 |  1  |  1  |
>
> All variants adopt the same total number of RL training steps=500. Even if the CoT is not fully shortened, training stops at step 500. The results on Qwen2.5-7B are:
>
> | Step | GSM8K | MATH | AIME | AMC  | Minerva | Olympiad | Avg  |
> | ---- | ----- | ---- | ---- | ---- | ------- | -------- | ---- |
> | 1    | 91.4  | 76.5 | 20.0 | 60.0 | 37.1    | 35.9     | 53.5 |
> | 2    | 90.8  | 77.1 | 16.7 | 52.5 | 38.6    | 38.7     | 52.4 |
> | 3    | 91.6  | 77.2 | 13.3 | 55.0 | 31.6    | 36.3     | 50.8 |
>
> A larger shortening step makes each stage substantially harder and disrupts the progression. Therefore, we set the step size to −1.
> This ablation will be added to the revision.
>
> ## **For W4 (Formatting issues)**
>
> Thank you for pointing this out. We will correct the citation commands and revise the notation to remove ambiguity. All formatting issues will be fixed accordingly.
>
> ## **For W5 (Experiments)**
>
> Thank you for the comment.
>
> First, we clarify that Table 3 compares **final results** trained on **the same base models using different methods**. Many open-source models in the table are trained with **much larger datasets than EvoCoT**. For example, “PRIME (380K)” indicates the size of its training set. In contrast, EvoCoT makes **more effective use of the available data** — it can solve challenging problems in GSM8K and MATH that other mthods fails to address. Considering that our training data **only includes GSM8K and MATH**, EvoCoT remains competitive with models such as PRIME and Open-Reasoner that rely on substantially broader datasets.
>
> Second, as noted in the Introduction, there are few prior works that **(1) do not use additional CoT distillation, (2) do not filter training data, and (3) apply curriculum learning.** Under these constraints, the current state-of-the-art curriculum baseline is SEC, which we adopt. EvoCoT achieves a significant improvement over SEC and other methods in the same category.
>
> The revised version will emphasize these distinctions and will annotate the training data scale for each model to make the comparison clearer.
>
> ---
>
> [1] Shibo Hao, Sainbayar Sukhbaatar, DiJia Su, Xian Li, Zhiting Hu, Jason Weston, Yuandong Tian: Training Large Language Models to Reason in a Continuous Latent Space. CoRR abs/2412.06769 (2024)
>
> [2] Hasan Abed Al Kader Hammoud, Kumail Alhamoud, Abed Hammoud, Elie Bou-Zeid, Marzyeh Ghassemi, Bernard Ghanem: Train Long, Think Short: Curriculum Learning for Efficient Reasoning. CoRR abs/2508.08940 (2025)
>
> [3] Xuezhi Wang, Denny Zhou: Chain-of-Thought Reasoning Without Prompting. NeurIPS 2024

---

### Author Response · Authors · 2025-11-22
**Author Summary Response**

We thank all reviewers for their constructive feedback. The comments mainly concern **(1) why EvoCoT works**, **(2) training overhead**, and **(3) experimental setup**. We summarize our responses below.

## 1. Why EvoCoT works
EvoCoT works because it enables the model to **use the available training data more effectively**, especially on the hard items that current models consistently fail to solve. For Qwen2.5-7B, **8.8% of GSM8K** and **22.0% of MATH** problems remain unsolved throughout training, leading the GRPO curve to stay at **0** on these cases in Figure 2. EvoCoT improves exploration under sparse verifiable rewards by allowing the model to self-learn from **multiple partially-completed reasoning states**. This expands the reachable solution space and helps the policy avoid the exploration bottlenecks observed in standard GRPO. Under this limited data regime, EvoCoT remains competitive with methods such as **PRIME** and **Open-Reasoner**, which rely on substantially larger and broader datasets, and achieves clear improvements over **SEC** and other curriculum-based approaches.

## 2. Training Overhead
Stage-1 is a **one-time preprocessing step** and is not part of RL training. During Stage-2, EvoCoT uses the same sampling procedure and on-policy GRPO updates as the baseline, so the per-step sampling cost is unchanged. Under identical configurations on Qwen2.5-7B, the total runtime increases only by **≈4% (413 min → 427 min)**, mainly due to filtering invalid rollouts on harder items. Under the same wall-clock budget, EvoCoT still outperforms GRPO, confirming that the improvement derives from **better use of exploration signals**, not increased computation.

## 3. Experimental Setup
We clarified the full algorithm in **Appendix A (Algorithm 1)** and will refine notation for readability. EvoCoT is **fully on-policy**, since only continuations generated by the current policy contribute to updates, and Stage-1 information is never used for gradients. There is **no answer leakage**, because verification signals are never exposed to the model. Variations across benchmarks are consistent with their reliance on multi-step reasoning. We will also add ablation analyses on shortening strategies and provide clearer illustrations in the revision.

In summary, we appreciate the reviewers’ feedback and will incorporate the requested clarifications and analyses to further improve the paper.

---

### Note · Authors · 2025-12-10

I have read and agree with the venue's withdrawal policy on behalf of myself and my co-authors.